# Phytochemical Constituents and Antioxidant Enzyme Activity Profiles of Different Barley (*Hordeum Vulgare* L.) Cultivars at Different Developmental Stages

**Li-Na Deng [1],\*, Gong-Neng Feng [1],\*, Yue Gao [1], Yu-Xiang Shen [1], Hong-Shan Li [1], Yue Gu [1] and Hai-Ye Luan [2]**

[1] College of Marine and Biology Engineering, Yancheng Institute of Technology, Yancheng 224051, China; gaoyue66668@163.com (Y.G.); ycshenyuxiang@163.com (Y.-X.S.); lhs158@126.com (H.-S.L.); yuegurugao@163.com (Y.G.)

[2] Institute of Agricultural Sciences in the Coastal Area in Jiangsu, Yancheng 224002, China; luanhaiye@163.com

\* Correspondence: dln@ycit.cn (L.-N.D.); ffyalce@ycit.cn (G.-N.F.)

**Abstract:** Barley grass possesses high nutritional value and antioxidant properties. In this study, the phytochemical constituents and antioxidant enzyme activities in six cultivars of barley grass were explored at three developmental stages: tillering, jointing, and booting stages. Total chlorophyll (Chl *t*) and carotenoid (Car) content, chlorophyll *a/b* (Chl *a/b*) ratio, total nitrogen nutrition (TNN), and total soluble protein (TSP) content, and superoxide dismutase (SOD), peroxidase (POD), and polyphenol oxidase (PPO) activities were assayed. The results indicated that the cultivar × development interaction was significant and that developmental stage was the main factor affecting the parameters studied. Cultivars had a negligible effect on these parameters, which varied with the developmental stages. In the tillering stage, Chl *t* and Car content, TNN, and POD activity achieved their highest value; in the jointing stage, SOD activity peaked; in the booting stage, Chl *a/b* ratio, TSP content, and PPO activity showed their highest values. TNN showed a negative correlation with TSP. Compared with those in the jointing, Chl *t*, Car, TSP, TNN content, Chl *a/b* ratio, and POD and PPO activities increased in the booting and the tillering stages, whereas SOD activity decreased. The differences in phytochemical constituents and antioxidant enzyme activities in barley grass were mainly correlated with the developmental stages. The aim of this study was to demonstrate the influence of developmental stages of barley grass on its phytochemical profile and antioxidant activities. Our results will help understand the mechanism of action of barley grass and provide theoretical support for the therapeutic application of barley grass.

**Keywords:** barley grass; chlorophyll; carotenoid; protein; antioxidant activity

## 1. Introduction

Commonly known as barley, *(Hordeum vulgare* L.) belongs to the Poaceae family and is the one of the main cereals grown worldwide [1]. Although barley has been used as a cereal grain since ancient times, the nutritional value of barley grass has been overlooked. However, it is now being increasingly considered to possess high nutritional value and antioxidant properties and thus, draws increasing attention due to the health benefits associated with it [1]. Indeed, barley grass is currently considered a health product, as many studies have demonstrated its potential in treating human chronic diseases [2–5].

Barley grass has been shown to contain high concentrations of amino acids, pigments, antioxidants, and enzymes [4]. Proteins have a major role in the growth and maintenance of the human body,

and barley grass contains 20 amino acids involved in energy production, and cell building and regeneration [3]. Furthermore, barley grass is rich in pigments, which mainly include chlorophyll (Chl) and carotenoids (Car) [2]. The Chl *t* content in barley grass is sufficiently high to exert potential health benefits in humans; thus, a daily intake of 100–300 mg of Chl *t* has been demonstrated to be beneficial for the treatment of various diseases in humans, including cancer [2]. Pigments are reportedly associated with health-promoting effects, such as antioxidant potential [2]. Significant positive correlations between total Chl *t* and Car content and total phenolics, and between total phenolics and antioxidant potential have also been reported [2,6–8]. Recently, synthetic antioxidants have started to be widely used in the pharmaceuticals, cosmetics, and processed food industries. However, their long-term use is accompanied by toxic and other adverse effects, raising concern over the use of natural antioxidants. Natural antioxidant chemicals in barley grass include not only folate, phenolics, and flavonoids, but also antioxidant enzymes, which have significant therapeutic potential for patients with cancer and diabetes [3].

The phytochemical and biochemical profiles of barley grass vary significantly, depending on the cultivar, season, location, and environmental conditions including, pH, light intensity and spectral quality, and temperatures [8,9]. Exploring the changes in phytochemical profiles and antioxidant enzyme activities among different cultivars at different growth stages is of great importance for the study of the mechanisms of action underlying the beneficial effects of barley grass on human health. Moreover, it is not clear how the developmental stage or cultivar might influence the phytochemical composition and antioxidant activity of barley grass. Therefore, in the present study the changes in the nutritional status and antioxidant enzyme activities of six barley cultivars at different developmental stages were explored. Thus (1) we evaluated Chl *t* and Car content, chlorophyll *a/b* (Chl *a/b*) ratio, total nitrogen nutrition (TNN), total soluble protein (TSP) content, and antioxidant enzyme activities including superoxide dismutase (SOD), peroxidase (POD), and polyphenol oxidase (PPO); (2) we performed correlation analysis of barley grass phytochemical constituents and antioxidant enzyme activities; and (3) we carried out principal component analysis (PCA) to analyze the effects of cultivar × development interactions on quality parameters.

## 2. Materials and Methods

### 2.1. Plant Materials

Six barley cultivars were used in this study. The seeds of Supi3, Supi6, Supi7, Supi11, and Supi12 were obtained from the Jiangsu Coastal Area Institute of Agricultural Sciences; and the seed of hua30 was obtained from Jiaxing Municipal Academy of Agricultural Sciences. All seeds were sown in the Jiangsu Coastal Area Institute of Agricultural Sciences (33°35′ N, 120°15′ E) on 3 November 2018. Jiangsu Coastal Area Institute of Agricultural Sciences is located in Yancheng, north of Jiangsu Province, China. It experiences a subtropical monsoon climate, which is characterised by rich rainfall and abundant sunshine. The first cutting was done on 6 January 2019 (tillering stage); the second cutting was done on 4 March 2019 (jointing stage); the third cutting was done on 5 April 2019 (booting stage). Barley grass samples were randomly selected at each developmental stage in the field. The fresh cutting grass from each cultivar was stored at 4 °C and returned to the laboratory for immediate testing. The barley grass leaves were excised and used for the analytical determinations described below.

### 2.2. Chlorophyll and Carotenoid Content in Barley Grass

A fresh weight (0.2 g FW) of barley was grounded using a mortar and pestle with ice-cold 95% (*v/v*) ethanol to extract pigments. The extract was centrifuged at 6000× *g* for 10 min, and the supernatants were pooled and diluted up to 10 mL with 95% (*v/v*) ethanol. The absorbance of the sample was recorded at 470, 665, and 649 nm (Thermo Fisher Scientific, Waltham, MA, USA). The Chl *a*, Chl *b* and total Car content was calculated using to the formulae of Lichtenthaler et al. [10].

### 2.3. Analysis of Total Nitrogen Nutrition in Barley Grass

Fresh barley grass samples were oven-dried in an oven at 75 °C to a constant weight, and then ground into a powder. The powdery sample (0.3 g) was placed in a digestion tube, and then potassium sulfate/copper sulfate (KjelTab) (6.2 g) and concentrated sulfuric acid (20 mL) were added. The tube was placed in a Kjeldahl digestor (C. Gerhardt Analytical Systems, Guangdong, China) and heated until the sample mixture appeared clear blue, and then the digestion process was continued by heating the sample for 30 min. The digestion residue obtained was cooled and diluted up to 100 mL with nanopure water. The digestion solution (10 mL) was placed in a distillation unit and distilled after adding sodium hydroxide (10 mL, 10 M). Distilled ammonia was adsorbed on 20 mL of boric acid (0.3 M) for 30 min. The collected solution was then titrated with hydrochloric acid (0.01 M). Total nitrogen content was measured using the Kjeldahl method [11].

### 2.4. Measurement of Soluble Protein in Barley Grass

Soluble protein content was measured using Coomassie Brilliant Blue G-250 (Solarbio Life Sciences, Beijing, China) [12]. Fresh barley grass samples (0.5 g) were homogenized with nanopure water (10 mL) in a mortar and pestle kept on ice and then centrifuged at $3000\times g$ for 10 min. Brilliant Blue G-250 (5 mL) was added to 1 mL of sample solution and the absorbance was measured at 595 nm after 5 min of incubation at room temperature.

### 2.5. Determination of Superoxide Dismutase (SOD) Activity

The SOD activity was measured using the xanthine oxidase (hydroxylamine) method. Fresh barley grass samples (0.2 g) were homogenized with 5 mL of phosphate buffered saline (PBS, pH = 7.2) in a mortar and pestle kept on ice and then centrifuged at $4000\times g$ for 20 min. The supernatant was collected and SOD activity was determined using a SOD assay kit (Nanjing Jiancheng Bioengineering Institute, Nanjing, China), according to the manufacturer's instructions. Sample absorbance of the sample was measured at 550 nm. The change in absorbance of 1 per milligram of tissue to produce 50% inhibition of reduction of nitrite in 1 mL reaction solution was considered one enzyme activity unit (U).

### 2.6. Determination of Peroxidase (POD) Activity

The POD activity was measured by the decomposition of $H_2O_2$ into water and molecular oxygen as determined by the peroxidase assay kit (Nanjing Jiancheng Bioengineering Institute, Nanjing, China), according to the manufacturer's instructions. Fresh barley grass samples (0.2 g) of barley were homogenized with PBS (9 mL) in a mortar and pestle kept on ice and centrifuged at $3500\times g$ for 10 min. Then, the supernatant was collected and used as enzyme extract. Double distilled water was used as control. Absorbance of the reaction mixture was measured at 430 nm (3 min). The change of absorbance of 1 per milligram of tissue per minute in the reaction system was considered one enzyme activity unit.

### 2.7. Determination of Polyphenol Oxidase (PPO) Activity

The PPO activity was determined using the polyphenol oxidase assay kit (Nanjing Jiancheng Bioengineering Institute, Nanjing, China). Fresh barley grass samples (0.1 g) of barley were homogenized with phosphate-buffered saline (PBS, 1 mL) in a mortar and pestle kept on ice, and then 1 mL of precooled enzyme-extraction solution was added. The sample mixture was centrifuged at $8000 \times g$ for 10 min at 4 °C, and the supernatant was collected and used as the enzyme extract (all operations were performed on the ice). PBS (0.6 mL) and enzyme extract (0.15 mL) were added into a test tube containing pyrocatechol solution (0.15 mL); 0.15 mL of boiled enzyme solution was used as control. The sample was shaken well for thorough mixing and then placed in a water bath at 37 °C for 10 min. Then, the sample mixture was immediately placed in a water bath at above 90 °C for 5 min, allowed to stand until cooling and centrifuged at $10,000\times g$ for 10 min. The absorbance of the sample

was measured at 420 nm (3 min). The change in absorbance of 0.01 per gram of tissue per minute in the reaction system was considered one enzyme activity unit.

## 2.8. Statistical Analyses

Statistical analyses all assays were performed in triplicates. The results are shown mean and standard deviation (SD) of three independent experiments calculated using Microsoft Excel. The statistical significance and correlations were determined using SPSS Statistics 17.0 software (IBM China Company Ltd., Beijing, China) to perform the paired *t*-tests. Asterisks indicate statistically significant differences (** $p < 0.01$, * $p < 0.05$, Student's *t*-test). Finally, the level of correlation among chemical constituent traits was analyzed by PCA using SPSS.

## 3. Results

### 3.1. Total Chlorophyll and Carotenoid Content in Barley Grass

The Chl *t* and Car content in six barley cultivars at different developmental stages is shown in Tables 1 and 2, respectively. There was no significant difference between the cultivars ($p > 0.05$). The Chl *t* and Car content in barley showed a consistent trend at different growth stages; their content decreased from the tillering to the jointing stages, and then increased in the booting stage. The Chl *t* and Car content in the six barley cultivars was the statistically highest in the tillering stage ($p < 0.05$). The Chl *t* content showed a significant reduction in the jointing stage, and the lowest Car content was also observed in the jointing stage ($p < 0.01$).

**Table 1.** Total chlorophyll (Chl *t*) content (mean ± standard deviation, SD) in the six barley cultivars at different developmental stages.

| Growth Stages | Chl *t* Content (Mg/100 g FW) | | | | | |
| --- | --- | --- | --- | --- | --- | --- |
| | Supi3 | Supi6 | Supi7 | Supi11 | Hua30 | Supi12 |
| Tillering stage | 79.84 ± 2.01 | 55.71 ± 2.41 | 20.39 ± 3.76 | 61.98 ± 2.84 | 54.71 ± 3.25 | 67.46 ± 1.67 |
| Jointing stage | 17.58 ± 5.79 | 11.01 ± 4.87 | 13.04 ± 0.03 | 11.75 ± 0.75 | 11.09 ± 0.63 | 15.94 ± 4.21 |
| Booting stage | 32.72 ± 1.05 | 23.94 ± 1.22 | 13.18 ± 0.22 | 44.29 ± 5.20 | 31.78 ± 1.76 | 46.41 ± 1.98 |

**Table 2.** The carotenoid (Car) content (mean ± SD) in the six barley cultivars at different developmental stages.

| Growth Stages | Car Content (Mg/100 g FW) | | | | | |
| --- | --- | --- | --- | --- | --- | --- |
| | Supi3 | Supi6 | Supi7 | Supi11 | Hua30 | Supi12 |
| Tillering stage | 11.98 ± 0.40 | 10.50 ± 1.78 | 4.74 ± 0.79 | 14.10 ± 0.83 | 10.09 ± 2.40 | 13.50 ± 2.59 |
| Jointing stage | 2.70 ± 0.26 | 2.13 ± 0.10 | 1.30 ± 0.55 | 2.87 ± 0.68 | 3.45 ± 0.19 | 4.90 ± 0.78 |
| Booting stage | 9.11 ± 0.32 | 8.03 ± 0.29 | 5.72 ± 0.53 | 11.57 ± 0.36 | 7.92 ± 0.18 | 10.68 ± 2.84 |

### 3.2. Chlorophyll a/b Ratio in Barley Grass

The Chl *a/b* ratio in the six barley cultivars showed the same trend as the Chl *t* and Car content at different developmental stages (Table 3), with no significant difference among the cultivars ($p > 0.05$); however, the values for Chl *a/b* ratio were significantly different at different developmental stages ($p < 0.01$), initially decreasing from the tillering to the jointing and then increasing in the booting stage. The maximum value for the Chl *a/b* ratio was recorded in the booting stage.

### 3.3. Total Nitrogen Nutrition (TNN) and Total Soluble Protein (TSP) Content in Barley Grass

The TNN and TSP content in the six barley cultivars at different developmental stages were determined. There was no significant difference among the cultivars ($p > 0.05$). The TNN in the six barley cultivars reached the peak in the tillering stage (Figure 1), while the mean value for TNN was 229.90 mg/g DW ($p < 0.01$). There was no significant difference in TNN between the jointing and

booting stages ($p > 0.05$). TNN decreased in the jointing stage and then slightly increased in the booting stage; meanwhile, the mean TNN value in the jointing and booting stages was 163.70 and 173.22 mg/g DW, respectively. This trend was consistent with that described for Chl and Car content.

**Table 3.** Chlorophyll a/b (Chl *a/b*) ratio (mean ± SD) in the six barley cultivars at different developmental stages.

| Growth Stages | Chl a/b Ratio | | | | | |
|---|---|---|---|---|---|---|
| | Supi3 | Supi6 | Supi7 | Supi11 | Hua30 | Supi12 |
| Tillering stage | 2.82 ± 0.04 | 2.78 ± 0.06 | 1.75 ± 0.15 | 2.67 ± 0.14 | 2.44 ± 0.31 | 2.61 ± 0.13 |
| Jointing stage | 0.95 ± 0.05 | 0.77 ± 0.27 | 0.68 ± 0.18 | 1.28 ± 0.65 | 1.44 ± 0.33 | 2.02 ± 0.39 |
| Booting stage | 3.41 ± 0.65 | 2.93 ± 0.25 | 2.47 ± 0.81 | 2.77 ± 0.25 | 2.95 ± 0.42 | 3.05 ± 0.28 |

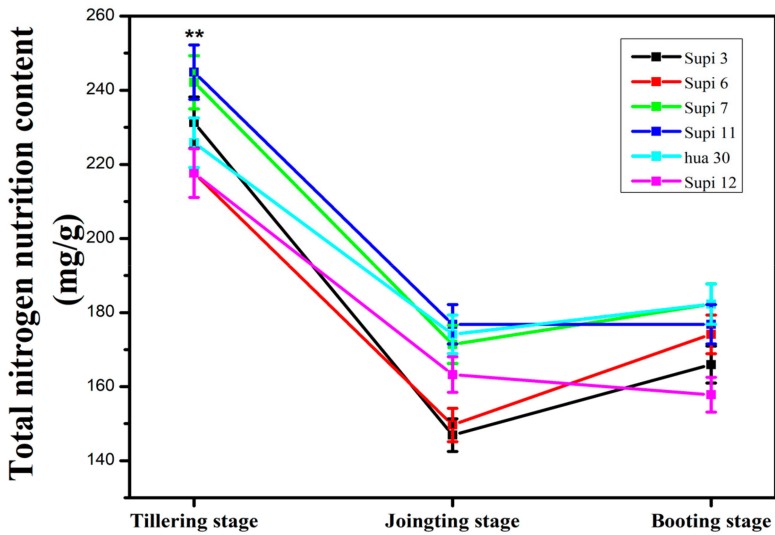

**Figure 1.** Changes of total nitrogen nutrition (TNN) content in the six barley cultivars at different developmental stages. Data represent mean values standard deviation (SD) from three independent experiments. Bars show standard deviations of the mean. Asterisks on top of the bars indicate statistically significant differences (** $p < 0.01$, Student's *t*-test).

A good linear relationship was observed in a standard curve of soluble protein of bovine serum, with correlation coefficients of 0.9921, 0.9971, and 0.9940, in the tillering, jointing, and booting stages, respectively (Figure 2A). The TSP content was increased with growth (Figure 2B). The mean TSP content increased marginally and ranged from 16.45 (jointing stage) to 21.74 (tillering stage) mg/g FW ($p > 0.05$); The TSP content reached the peak in the booting stage, and the mean TSP content was 55.10 mg/g FW. The TSP content in the booting stage was significantly higher in the booting than in either tillering or jointing ($p < 0.01$). The TNN and TSP content showed opposite patterns at different developmental stages.

### 3.4. Antioxidant Enzyme Activities in Barley Grass

The SOD, POD, and PPO activities are shown in Figure 3; obvious differences in the antioxidant enzyme activities were observed at different developmental stages of barley grass. However, there was no significant difference among the cultivars ($p > 0.05$), each cultivar showing a consistent trend at different growth stages. Thus, SOD activity was the highest in the jointing stage and the lowest in the booting stage. Further, SOD activity was significantly different ($p < 0.01$) in the tillering, jointing, and booting stages. In turn, the activity of POD was the significantly lowest ($p < 0.01$) in the booting stage, whereas PPO activity was the highest ($p < 0.01$). POD and PPO activities in the tillering and jointing stages were not significantly different ($p > 0.05$).

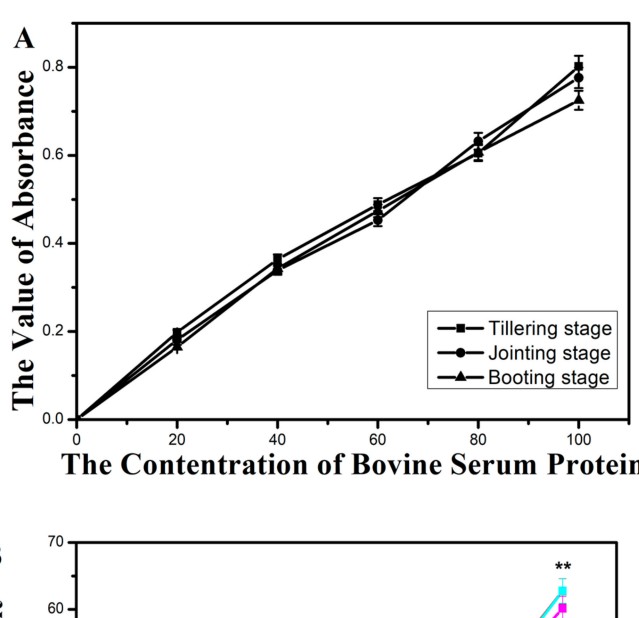

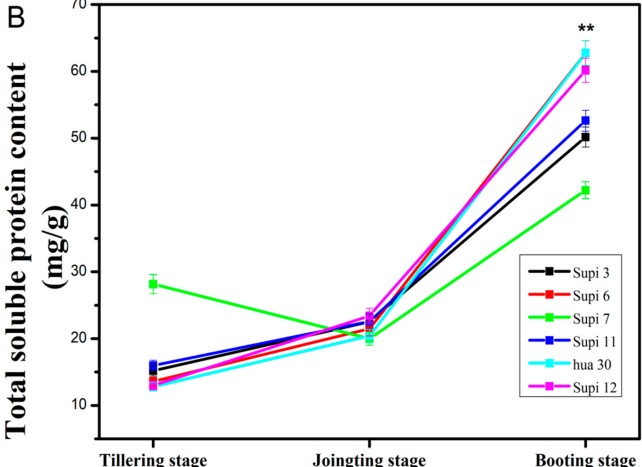

**Figure 2.** Standard curve of soluble protein of bovine serum at different developmental stages (**A**) and the total soluble proteins (TSP) in the six barley cultivars at different developmental stages (**B**). Data represent mean values standard deviation (SD) from three independent experiments. Bars show standard deviations of the mean. Asterisks on top of the bars indicate statistically significant differences (**\*\*** $p < 0.01$, Student's *t*-test).

### 3.5. Correlations Among the Quality Parameters

Correlations among the Chl *t*, and Car content, Chl *a/b* ratio, TNN, and TSP content, and SOD, POD, and PPO activities in barley grass are shown in Table 4. The Car content showed a significantly positive correlation with the Chl *t* content and Chl *a/b* ratio, with very strong correlation coefficients (0.90, 0.83, $0.8 < r < 1$; $p < 0.01$), respectively. TNN content showed a weak, negative correlation with the TSP content. In turn, SOD and POD activities showed moderately negative and significant correlations with the PPO activity, while POD activity showed a moderate, positive, and significant correlation with the SOD activity (0.40, $0.4 < r < 0.6$; $p < 0.05$).

There was no significant difference in the Chl *t* or Car content, Chl *a/b* ratio, TNN, and TSP content, or SOD, POD, or PPO activities among the six cultivars under study ($p > 0.05$; Table 5). However, all these parameters were highly significantly different at different development stages ($p < 0.01$). The cultivar × development interaction also showed a significant difference in the above mentioned parameters ($p < 0.05$).

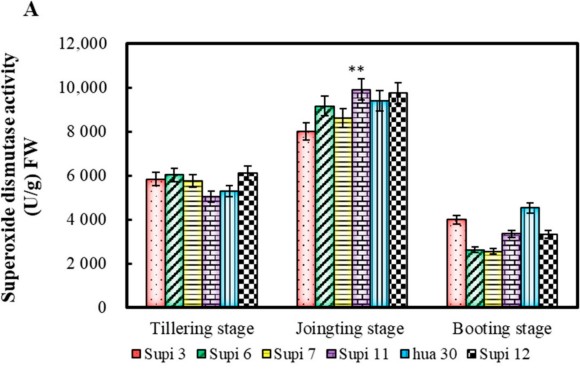

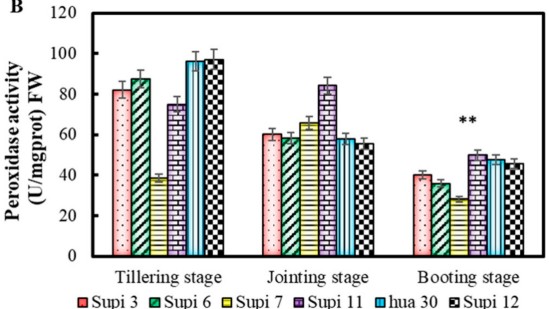

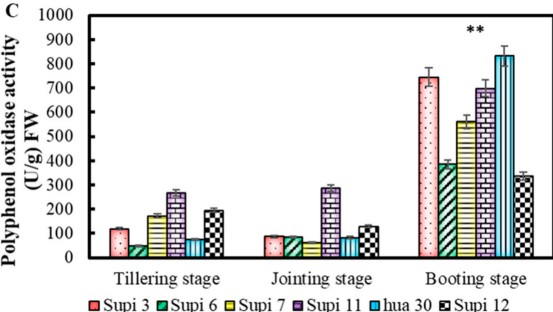

**Figure 3.** Changes of the superoxide dismutase (SOD) activity (**A**), peroxidase (POD) activity (**B**), and polyphenol oxidase (PPO) activity (**C**) in the six barley cultivars at different developmental stages. Data represent mean values standard deviation (SD) from three independent experiments. Bars show standard deviations of the mean. Asterisks on top of the bars indicate statistically significant differences (** $p < 0.01$).

**Table 4.** Correlation among the Chl *t*, and Car content, Chl *a/b* ratio, TNN, and TSP content, and SOD, POD, and PPO activities.

| Items | Chl *t* | Car | Chl *a/b* | TNN | TSP | SOD | POD | PPO |
|---|---|---|---|---|---|---|---|---|
| Chl *t* | 1.00 | | | | | | | |
| Car | 0.91 ** | 1.00 | | | | | | |
| Chl *a/b* | 0.63 ** | 0.83 ** | 1.00 | | | | | |
| TNN | 0.66 ** | 0.57 ** | 0.32 | 1.00 | | | | |
| TSP | 0.20 $^n$ | 0.09 | 0.49 * | 0.43 $^{n,*}$ | 1.00 | | | |
| SOD | 0.40 $^{n,*}$ | 0.63 $^{n,**}$ | 0.80 $^{n,**}$ | 0.22 $^n$ | 0.65 $^{n,**}$ | 1.00 | | |
| POD | 0.56 ** | 0.30 | 0.11 $^n$ | 0.46 * | 0.76 $^{n,**}$ | 0.40 * | 1.00 | |
| PPO | 0.29 $^n$ | 0.26 | 0.58 ** | 0.20 $^n$ | 0.81 ** | 0.64 $^{n,**}$ | 0.55 $^{n,**}$ | 1.00 |

$^n$ showed negative correlation, ** Correlation is significant at the 0.01 level (2-tailed), * Correlation is significant at 0.05 level (2-tailed).

**Table 5.** The analysis of variance (ANOVA) results (*p* value) of cultivar × development on selected composition traits of barley grass.

| Items | Chl *t* | Car | Chl *a/b* | TNN | TSP | SOD | POD | PPO |
|---|---|---|---|---|---|---|---|---|
| Cultivar | 0.056 | 0.033 | 0.352 | 0.586 | 0.999 | 0.989 | 0.169 | 0.408 |
| Developmental stage (D) | <0.001 | <0.001 | <0.001 | <0.001 | <0.001 | <0.001 | <0.001 | <0.001 |
| C × D | <0.001 | <0.001 | 0.032 | 0.041 | <0.001 | 0.007 | <0.001 | <0.001 |

*3.6. Principal Component Analysis (PCA) for the Cultivar × Development Interactions for Quality Parameters*

PCA was conducted to elucidate on quality parameters of six barley cultivars at three developmental stages (tillering, jointing, and booting) (Figure 4). The first PCA axis accounted for 43.31% of the overall variance and correlated with the phytochemical constituents. The second axis (42.65% of the variance) was associated with antioxidant potential. The Chl *t* and Car content, Chl *a/b* ratio, TNN content, and POD and PPO activities were positively associated with the first PCA axis, whereas SOD activity and TSP content were negatively associated with the first PCA axis. As observed in Figure 4, the three groups were clearly separated at different developmental stages. This further points to the trend that, compared with that in the jointing stage, the Chl *t* and Car content, Chl *a/b* ratio, TSP and TNN content, and POD, and PPO activities increased in the booting and tillering stages, whereas the SOD activity decreased. This indicates that developmental stage was the main factor affecting the quality of phytochemical constituents and antioxidant potential of barley grass.

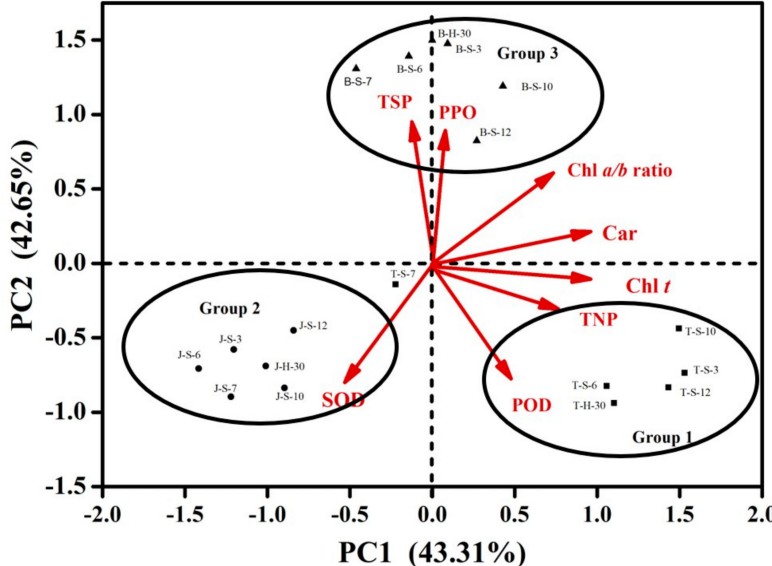

**Figure 4.** Principal component analysis (PCA) of the cultivar × development interactions in six barley cultivars at three developmental stages. PC1 accounted for 43.31% of variance and PC2 accounted for 42.65% of variance. T—tillering stage, J—joingting stage, B—booting stage.

## 4. Discussion

Recently, the use of barley grass has increased because of growing consumer interest in healthier food products. The higher nutritional values and health benefits of barley grass are attributed to the high concentrations of pigments and antioxidant substances. Nevertheless, the increasing global popularity of barley grass is due to its rich pigments profile. The Chl *t* pigments are strong natural antioxidants acting as free radical scavengers [13], and the Car pigments are associated with age-related eye diseases [14]. Studies have revealed that the Chl *t* and Car content in barley varies with the species, cultivars, cultivation pattern, and temperature [15]. In this study, six barley cultivars were grown under the same conditions, thus excluding the influence of the environment and other factors. The results indicated that the content of Chl *t* and Car varied with the developmental stage, and there was no

significant difference among the cultivars. The Chl *t* and Car content in barley grass was relatively high in the tillering and booting stages. Furthermore, with an increase in the Chl *t* content, there was a simultaneous increase in the Car content; this result is consistent with that of Ndukwe et al. [16]. The content of pigments is the highest during cell biosynthesis and energy consumption [2]. In the present study, the Chl *t* and Car content in barley grass was the lowest in the jointing stage after peaking in the tillering stage, which possibly suggests the rapid production of phenolics and other antioxidants in barley microgreens [8,17]. With an increase in the content of allelopathic phenolics, the biosynthesis of Chl *t* is reduced or even inhibited [18]. Compared with that in the tillering stage, the Chl *a/b* ratio was high in the booting stage, and Chl *b* was more resistant to degradation under high temperature treatment [19]. This study was carried out for the comprehensive profiling of Chl *t* and Car in six barley cultivars at three different developmental stages.

Proteins determine the quality of barley, as they are a significant nutritional component for plant growth and human consumption [20]. In the present study, we found that the TNN content was higher than the TSP content in the three developmental stages. The Kjeldahl method is used to determine not only the protein component but also the relative nitrogen component, such as nitrate, ammonia, urea, nucleic acids, free amino acids, chlorophylls and alkaloids, all of which contain nitrogen [11]. The Bradford method is routinely used to determine the soluble protein content [21]. In the present study, the TNN content showed a weak negative correlation with the TSP content, and the TNN and TSP contents reached a peak in the tillering and booting stages, respectively. We speculated that the accumulation of TNN might be influenced by the Chl *t* content in barley grass. The increase of TSP in booting stage may result from the synthesis of highly proteins for reproductive development [22].

In this study, the enzyme activities varied with the different developmental stage. The SOD activity showed a positive correlation with the POD activity; both these enzyme activities showed a negative correlation with the PPO activity. Superoxide dismutase is an important antioxidant enzyme that eliminates free oxygen radicals in cells, and POD plays an important role in enhancing antioxidant capacity, which promotes the oxidation of phenolics [23]. Barley grass showed relatively high SOD and POD activities in the jointing stage. This suggests that barley grows rapidly in the jointing stage, during which free radical production and defensive responses actively occur. Consequently, high levels of antioxidant enzymes are required, resulting in very high mitochondrial activities and superoxide production [24]. The relative decrease in the SOD and POD activities during the booting stage might be due to the metabolic conversion, decomposition, and inactivation of enzymes. In addition, we found that the highest PPO activity occurred in the booting stage. Polyphenol oxidase is a key enzyme that is involved in the lignification of the plant cell wall, wherein it participates in the process of tissue aging [25]. The PCA is an unsupervised method for classifying sample groups based on the inherent similarities or dissimilarities in the chemical composition data without prior knowledge of sample classes [26]. In the present study, the three developmental stage groups were clearly separated by the two principal component axes, PC1 and PC2, which suggested that the compounds in barley grasses at three developmental stages could be clearly distinguished. The changes in the antioxidant enzyme activities among the different developmental stages are attributable to the transformation of chemical substances and the interaction of antioxidant enzymes [23]. The metabolism and antioxidant activity have been determined by phytochemical profiles and the enzyme activity through complex interactions at the different developmental stages of barley grasses, rather than simply being caused by the change of certain material [23]. However, the mechanisms underlying the antioxidant activity of the chemical constituents and enzymes isolated from barley grass need to be further investigated.

## 5. Conclusions

In conclusion, our results showed that chemical constituents and antioxidant enzyme activities in barley grass differed significantly during different developmental stages, whereas cultivars had much less of an effect on such constituents. The Chl *t* and Car content, and TNN and POD activity were the highest in the tillering stage; in turn, the SOD activity was the highest in the jointing stage; and

the Chl *a/b* ratio, TSP content and PPO activity were the highest in the booting stage. The changes in antioxidant activity during the three key developmental stages of barley grass are attributable to the transformation of chemical substances and to the interaction of antioxidant enzymes. The findings summarized herein contribute to expanding our understanding of the changes in phytochemical profiles and antioxidant activity profiles in barley grass. Overall, this will foster the understanding of the mechanisms of action of chemical constituents and antioxidant enzyme activities isolated from barley grass, and thus provide sound theoretical support for the therapeutic application of barley grass in the treatment of certain chronic diseases that affect humans.

**Author Contributions:** Conceptualization, Methodology, Writing-review and editing, L.-N.D.; Methodology, Resources, G.-N.F., Y.-X.S. and H.-S.L.; Methodology, Data curation, Formal analysis, Y.G. (Yue Gao) and Y.G. (Yue Gu); Resources, Funding acquisition, H.-Y.L. All the authors approved the version to be published. All authors have read and agreed to the published version of the manuscript.

**Funding:** This work was financially supported by the policy guidance plan of Jiangsu province (subei science and technology special project) (No. SZ-YC2018013).

**Conflicts of Interest:** The authors declare no conflict of interest.

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
