# Peer review of "Phytochemical Constituents and Antioxidant Enzyme Activity Profiles of Different Barley (Hordeum Vulgare L.) Cultivars at Different Developmental Stages"

_agronomy, doi:10.3390/agronomy10010037_

Round 1

Reviewer 1 Report

Although this manuscript does not provide any new ideas, it is well written and clear. Only some general languages mistakes and some comments need to be fixed

The abstract should include the aim of the study. Please state the aim of the study clearly in this section

Line 13 and Line 63 : Please avoid the personal pronouns (we)

Line 51-55: The sentence is too long 

The discussion part in general needs to be improved

Line 288 - 29o The changes in antioxidant activity..... This part needs support. The author should provide citations

In the conclusion section, the author in line 301-302 stated that Overall this will foster the understanding of the mechanisms of action .....( please explain this part clearly in the discussion)  

Reviewer 2 Report

Overall the papers is interesting and provides some new data on barley composition at different stages. However, more state-of the art methodology could have been applied to obtain more interesting data such as chromatography and mass spectrometry for carotenoid and chlorophyll composition.

In addition, I have some points I would ask the authors to clarify.

1) Could you please describe the differences within genotypes? What does the genotype code mean?

2) What was the sample size of seeds for each genotype?
